# Intra- and inter-rater reliability, agreement, and minimal detectable change of the handheld dynamometer in individuals with symptomatic hip osteoarthritis

**Gilvan Ferreira Vaz**[1,2☯]*, **Felipe Florêncio Freire**[2☯], **Henrique Mansur Gonçalves**[3‡], **Marcus Alexandre Brito de Aviz**[4‡], **Wagner Rodrigues Martins**[1‡], **João Luiz Quagliotti Durigan**[1,5‡]

**1** Faculty of Ceilândia, Rehabilitation Sciences Program, University of Brasilia, Brasília, DF, Brazil, **2** Medicine Division, Department of Orthopaedics, Hospital das Forças Armadas (HFA), Brasília, DF, Brazil, **3** Orthopedic Department of the Santa Helena Hospital, Brasília, DF, Brazil, **4** Anesthesiology Department of the Institute Hospital de Base, Brasília, DF, Brazil, **5** Faculty of Ceilândia, Rehabilitation Sciences Program, Laboratory of Muscle and Tendon Plasticity, University of Brasilia, Brasília, DF, Brazil

☯ These authors contributed equally to this work.
‡ HMG, MABA, WRM and JLQD also contributed equally to this work.
* gilvanvaz@gmail.com

**Data Availability Statement:** Raw data was uploaded to the repository Figshare as

## Abstract

### Introduction

The handheld dynamometer has been validated to measure muscle strength in different muscle groups. However, to date, it has not been tested in individuals who experience pain induced by hip osteoarthritis. The current study aimed to evaluate the intra- and inter-rater reliability, agreement, and minimal detectable change of the Lafayette handheld dynamometer, model 1165, to assess the peak force (Pk) and average peak force (Af) of hip muscles in individuals with symptomatic hip osteoarthritis.

### Methods

Twenty participants with hip osteoarthritis (mean ± SD age: 58.7±15.3 years; body mass index: 28.8±4.2 kg/m²) and pain intensity on the Visual Analogue Scale ≥ 4 (8.05±1.2) were recruited to participate in this study. Pk and Af of hip flexors (seated position), abductors and adductors (supine position), and extensors (prone position) were collected in a single day by two independent raters, each one obtaining test and retest in randomly ordered separate sessions.

### Results

The intra-rater intraclass correlation coefficient (ICC) was classified as good (>0.75) or excellent (≥0.90) for all muscle groups and all inter-rater ICCs were classified as excellent. Rater A had a lower standard error of measurement compared to rater B, ranging from 0.15 to 0.58 kilogram-force (Kgf) compared with 0.34 to 1.25 kg, respectively. However, the inter-rater comparison showed a minimal detectable change (MDC) of < 10% for all Pk and Af

recommended and is available at: dx.doi.org/10.6084/m9.figshare.6025748.

**Funding:** This study was supported by FAPDF (Fundação de Apoio a Pesquisa do Distrito Federal), process number 1008, grant number 003/2023.

**Competing interests:** The authors have declared that no competing interests exist.

measures for hip adductors and extensors. Finally, the inter-rater Bland-Altman analysis demonstrated good agreement for abductors, adductors, and extensors.

## Conclusion

Despite pain and dysfunction related to hip osteoarthritis, the mean of two measures using a handheld dynamometer was shown to be a reliable tool to assess hip muscle strength, with good to excellent intra- and inter-rater ICCs, satisfactory agreement, and small values for MDC.

## Introduction

Hip Osteoarthritis (OA) is an end-stage disease from various causes, resulting in chronic hip pain, dysfunction, and stiffness. It is estimated that symptoms are present in 5 to 10% of adults older than 40 and 45 years, considering the Spanish and American populations, respectively, with a higher prevalence with increasing age [1, 2]. Chronic hip pain is associated with muscle atrophy and weakness, as demonstrated in a meta-analysis conducted with pooled data from thirteen articles. Collectively, the authors observed a reduction in muscle strength in individuals with osteoarthritis that mainly affects hip flexors (-22%) and hip extensors (-21%), or abductors (-31%) and adductors (-25%) compared to healthy control groups [3].

Muscle weakness and atrophy seem to have a central role in the dysfunction related to hip osteoarthritis, as demonstrated by imaging studies and isometric dynamometry [3–5]. Both are deeply connected to the degree of radiographic OA classification, and their progress should be avoided through participation in exercise programs that include aerobic and strengthening exercises [6, 7]. However, a reliable and easy method seems to be necessary to measure the strength of hip muscles in the clinical routine, in order to monitor disease and treatment progression in the rehabilitation process.

Measurement of Peak Force (Pk) has been considered the gold standard method for isokinetic test parameters to evaluate muscle function [8] and can be acquired with fixed or portable dynamometry. Handheld dynamometers (HHD) have been suggested as a practical, feasible, and simple tool to assess isometric lower limb muscle strength in the clinical setting [9] compared to fixed laboratory-based dynamometry, such as isokinetic dynamometers [10, 11]. In addition, manual dynamometers require little training for proficient application [10, 12] and have lower costs than fixed laboratory-based dynamometry [13, 14].

Several studies have validated and recommended the use of different HHDs to measure hip muscle strength, with good to excellent intraclass correlation coefficients (ICC). The validity of two HHDs compared with a fixed laboratory-based dynamometer was previously demonstrated [11], with good to excellent reliability, particularly for proximal muscle groups in the lower limbs of health subjects. Other author also found good to excellent reliability when evaluating hip flexors and adductors of young adult football players [15], or when evaluating young, healthy adults with similar results when testing hip and knee strength [16]. The literature assessed different hip muscles in various protocols and also recommended the use of manual devices to acquire hip muscle strength in healthy subjects [9, 17, 18]. Only one study tested the use of the HHD in older people (> 65 years old) and found good reliability for hip and knee muscle strength measures, without specifying lower limb articular disease [10]. There are no definitive findings about the reliability of HHD measurements in participants with symptomatic hip OA.

Accordingly, it is crucial to determine if pain intensity related to hip OA could affect the reliability of HHD in muscle strength evaluations of the hip, since comfort may be a potential limitation for strength performance [12]. Furthermore, the standard error of measurement (SEM) and minimal detectable change (MDC) need to be determined to allow comparability for routine measurements in clinical settings of symptomatic hip OA patients. The purpose of this methodological study was to analyze the reliability, agreement, and minimal detectable change of an HHD in individuals with chronic hip pain related to OA. We hypothesized that HHD could be a reliable tool to measure muscle strength for hip muscles even if symptomatic hip OA is present. Our findings could help clinicians and physical therapists to design more rational assessment strategies for individuals with chronic hip pain related to OA, using a tool that requires little training, is low cost, and minimizes the time needed by patients and clinicians.

## Methods

### Study design

A methodological study with repeated measures was conducted to determine the intra- and inter-rater reliability, agreement, and MDC for strength assessment of hip muscles obtained with the HHD, testing subjects who experience chronic hip pain. Participants were assessed in a single-day session. Data collection occurred between August 2021 and March 2022 after approval by the local Ethics Committee (CAAE 40347320.1.1001.0025), following the Helsinki Declaration of 1975. All participants signed an informed consent before data collection. The research was conducted at the Hospital das Forças Armadas (Brasília, Brazil) and Instituto Hospital de Base (Brasília, Brazil), following the guidelines for reporting reliability and agreement studies (GRRAS) [19].

### Participants

Twenty participants {40% female, mean age 58.7 (± 15.3) years, age range = 41–79 years, body mass index = 28.8 (± 4.2) kg/m$^2$} with symptomatic hip OA were enrolled in the present study from the Orthopedic Department of two tertiary hospitals. Eligibility and demographic data were obtained using an interview questionnaire formulated by the authors. Study procedures were explained to potential participants, and they were assigned to the study protocol if eligible and after giving written informed consent. Participants were included if they presented hip OA radiographically classified as type II (Definite osteophytes, possible joint space narrowing), III (moderate osteophytes, definite joint space narrowing, some sclerosis, possible bone-end deformity), or IV (Large osteophytes, marked joint space narrowing, severe sclerosis and definite bone ends deformity) according to the Kellgren and Lawrence classification [20, 21], performed by rater B. All participants had previously been screened with x-ray images as part of their usual care, and no additional image investigation was performed. Other causes of the reported pain, lower limb and back, were also excluded as the primary source of pain, and range of motion was tested to guarantee the hip as the source of the symptoms.

### Instruments

Pain intensity was assessed using the Visual Analogue Scale (VAS), with faces ranging from 0 to 10, presented to the participants at the eligibility interview and after each protocol sequence of muscle strength assessment [22, 23]. The Western Ontario and McMaster Universities Index (WOMAC) was used as a Health-related Quality of Life (HrQOL) questionnaire developed for patients with hip and knee OA as a self-reported tridimensional scale. The

questionnaire evaluates pain, function, and joint stiffness (five questions for the subscale of pain, two questions for the subscale of stiffness, and seventeen questions for the subscale of function). Answer options are presented on a 5-point Likert scale. The total possible score ranges from 0 to 96; the fewer points scored, the better the patient's HrQOL [24, 25]. Lastly, to characterize the severity of hip OA, the Harris Hip Score (HHS) was applied by one of the examiners (rater A) to evaluate four domains: Pain (0–44 points), function (0–47 points), absence of deformity (0 or 4 points), and mobility (0–5 points). Scores range from 0 to 100, with higher scores demonstrating less compromised hip joints [26–28].

Procedures

The HHD Lafayette Manual Muscle Testing System Model-01165 (Lafayette Instrument Company, Lafayette IN, USA) was used to assess hip muscle strength during a three-second maximal effort, following the protocol sequence: hip flexors (seated position), hip abductors, and adductors (supine, long-lever), and hip extensors (prone, long-lever) performed on a regular examination table and collected on the same day by both raters. This time frame was chosen considering the clinical context, that individuals with symptomatic hip OA and older adults have difficulties in moving around, which could affect adherence to a second day of evaluation. We assumed that patients would be more interested in participating in the study protocol if measurements were taken on the same day as their regular medical evaluation. In addition, our protocol aimed to mimic the clinical routine of physicians and physiotherapists when evaluating their patients, reproducing a more realistic scenario to be adopted in practice [19, 29].

Participant and rater positions have been described elsewhere [11], with some minor modifications. Hip flexors (Fig 1A) were evaluated with the participant on an examination table, seated with both legs hanging off the table and arms positioned at the sides of the body, and both knees and hips at 90˚. The assessor was placed right in front of the affected lower limb, holding the HHD with both hands at the anterior aspect of the thigh, 1 to 2 cm above the superior edge of the patella. Participants were instructed to push against the HHD, trying to flex the hip with the maximal force for three seconds. Hip abductors (Fig 1B) were tested in the supine position, hands crossed in front of the chest, hip and knee at 0˚, with the assessor

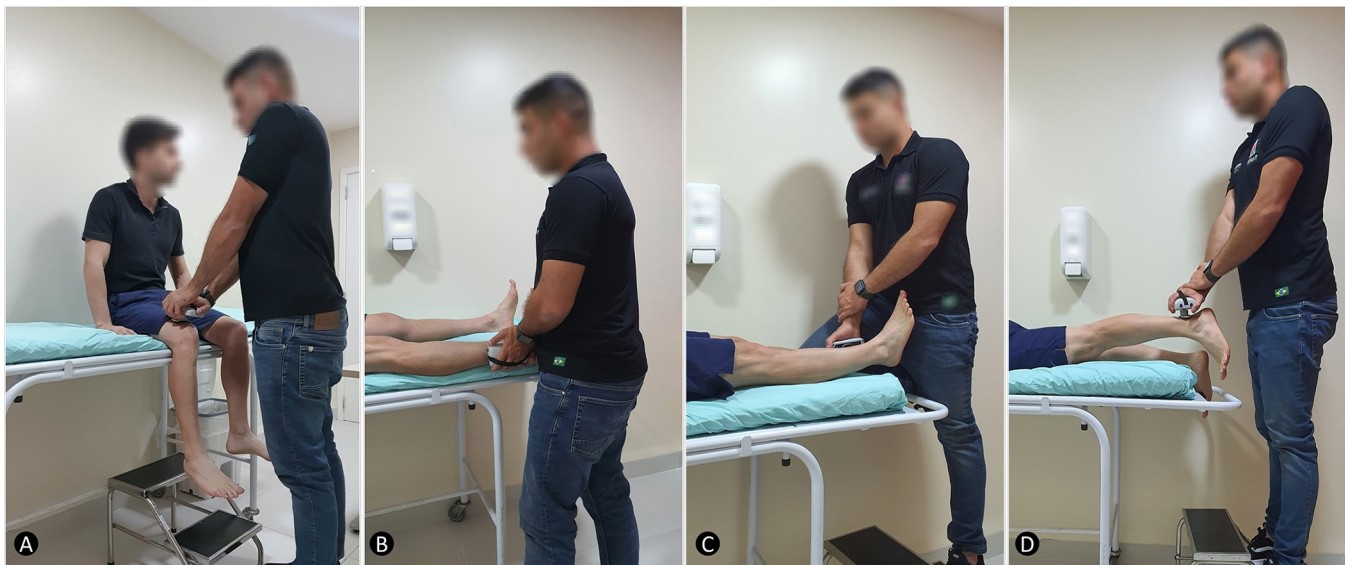

**Fig 1. Test positions for hip muscle strength assessment.** A) Hip flexors with the participant in the seated position. B) Hip abductors with the participant in the supine position. C) Hip adductors with the participant in the supine position. D) Hip extensors with the participant in the prone position.

standing by the side of the examination table and holding the HHD with both hands above the lateral malleolus (long-lever), using their own body to stabilize it. Similarly, the participant tried to abduct the affected hip against the HHD. Hip adductors (Fig 1C) were evaluated with the participant in the same position, but now, with the HHD held above the medial malleolus (long-lever) and the examiner placing their knee in the middle of both participants' ankles. In this situation, the participant was encouraged to adduct only the affected leg. Finally, the participant was instructed to lie in the prone position to evaluate hip extensors (Fig 1D), arms crossed under the forehead, hip and knee at 0°. The rater stood immediately in front of the end of the table, holding the device with both hands, elbows extended, at 3–4 cm above the posterior calcaneal tuberosity (long-lever), followed by an attempt to extend the hip while maintaining the knee at full extension. All participants were advised not to flex the knee during hip extension.

Before every protocol sequence of muscle strength assessment, participants were instructed to push against the HHD with their maximum force for three seconds and were reminded that the test starts as they push the HHD and hear a single sound alarm and finishes as they hear a double sound alarm. A submaximal strength test trial was performed in the seated position with the non-affected limb to familiarize the participant with how the device works and the sound alarm. One demonstration was also performed in the supine and prone position to clarify how the test could be performed if required [10, 11]. None of the participants had any previous familiarity with this device.

Two independent raters performed data collection, both physicians (V.G.F.; F.F.F) with no experience with the HHD. Raters were allowed to practice the measurements protocol sequence for four months. Data were registered using REDCap (Research Electronic Data Capture) electronic data capture tools hosted at Instituto Hospital de Base [30, 31]. Each rater repeated measurements twice on the same day. To minimize any possible effect of cumulative pain resulting from test-retest, the order of data collection was defined using a randomized sequence generated on the website sealedenvelope.com (proportion of 1:1, in blocks of four). Participants were allowed to rest between each protocol sequence until they felt comfortable to start the next round [14]. The VAS for pain was measured after each sequence. Participants were given continuous encouragement to push harder against the HHD to obtain maximal isometric force during the 3 seconds of each test [11, 14].

## Statistical analysis

Descriptive statistics were used to describe participants' sociodemographic characteristics. The Shapiro-Wilk test was performed to confirm the normal distribution of the data. The Paired t-test was used to compare VAS for pain intensity between intra- and inter-rater measures. Assessment of intra- and inter-rater reliability regarding Pk and Af measures was conducted using the ANOVA 2-way random model, with a Confidence Interval of 95% (95%CI), to compare test-retest measures for intra-rater analysis, and the mean of test-retest for inter-rater analysis. To categorize the reliability between repeated measures, we assessed the intraclass correlation coefficient (ICC $_{2,1}$), and the correlation between measures was classified as poor (ICC < 0.5), moderate (0.5 ≥ ICC < 0.75), good (0.75 ≥ ICC < 0.90), and excellent (ICC ≥ 0.90) [11, 32]. To define the presence of bias in the data and establish the Limit of agreement (LoA) between raters, mean values considering the two measures were plotted with a 95% CI using the Bland-Altman (BA) method [33, 34]. Absolute reliability was evaluated by calculating the Standard Error of Measurement (SEM) and percentage of values (SEM%), and Minimal Detectable Change (MDC) and percentage of values (MDC %) for a 95% CI were calculated considering the following equation: $SEM = \left(\sqrt{\frac{SStotal}{n-1}}\right) x \sqrt{(1 - ICC)}$ and $MDC =$

$[z\ score(95\%\ CI)]\ x\ SEM\ x\ \sqrt{2}$ [35, 36]. Statistical significance was assumed when $p < 0.05$. All statistical analyses were performed using SPSS version 25 (IBM Corp., Chicago, IL, USA), and the BA graphs were plotted by GraphPad Software (San Diego, CA, USA). The sample size was calculated using an acceptable ICC of 0.70, an expected ICC of 0.90, and assuming an $\alpha$ of 5% and power of 80%, with a drop-out rate of 10%, resulting in a minimal sample of 20 participants [37].

## Results

The demographic data of the sample are shown in Table 1. Approximately 85% of the participants presented a defined joint space reduction associated with sclerosis and moderate to severe osteophytes (types III/IV), representing the whole spectrum of substantial alterations in the x-ray related to osteoarthritis. Considering all daily activities during the week before inclusion in the study, the pain intensity (VAS; mean ± SD) was 8.05±1.2, 95%CI {7.47–8.62}, and together with an HHS score of 50.2±20.1 and a WOMAC score of 63.5±14.0, the data show considerable pain, dysfunction, and a reduction in quality-of-life related to hip OA.

A statistical difference in VAS (mean±SD, 95%CI) was observed for pain intensity after test and retest for rater A (Test: 6.11±2.96, {4.68–7.36}; Retest: 6.74±2.94, {5.32–8.15}; $p = 0.01$) that was not observed for rater B (Test: 6.42±2.52, {5.20–7.73}; 6.74±2.74, {5.73–8.04}; $p = 0.49$), or between raters when considering the mean VAS for the pain intensity after two measures (A test-retest: 6.55 ± 2.96, {5.12–7.98}; B test-retest: 6.73 ± 2.39, {5.58–7.89}; $p = 0.54$). Therefore, considering the imprecision related to a VAS of ± 20mm [38] and the minimal clinically important difference in pain for hip osteoarthritis of 24mm and 30mm regarding a baseline VAS interval of 50 – 65mm and >65mm, respectively [39, 40], our result did not reach a meaningful change for intra- or interrater VAS between trials.

Table 2 shows the mean ± SD values of test-retest Pk and Af, relative reliability expressed as $ICC_{2,1}$, absolute reliability expressed as SEM, and $MDC_{95}$ for the four major hip muscle groups, comparing intra- and inter-rater reliability.

**Table 1. Characteristics of participants.**

| Characteristic | Sample (n = 20) |
| --- | --- |
| **Age, mean (SD), y** | 58.7 ± 15.28 |
| **Sex (%)** | |
| Male | 12 (60) |
| Female | 8 (40) |
| **Radiographic disease severity (%)** | |
| KL II | 3 (15) |
| KL III | 6 (30) |
| KL IV | 11 (55) |
| **Symptoms (%)** | |
| 6m – 1y | 1 (5) |
| 1y – 2y | 9 (45) |
| 2y – 5y | 5 (25) |
| > 5y | 5 (25) |
| **Body mass index, mean (SD)** | 28.82 ± 4.23 |
| **VAS (0–10), mean (SD)** | 8.05 ± 1.23 |
| **HHS (0–100), mean (SD)** | 50.2 ± 20.1 |
| **WOMAC (0–96), mean (SD)** | 63.5 ± 14.0 |

SD: standard deviation; y: year; m: month; KL: Kellgren and Lawrence classification; VAS: visual analogue scale; HHS: Harris Hip Score; WOMAC: Western Ontario and McMaster Universities.

**Table 2. Handheld dynamometer reliability analysis for hip muscle groups.**

| Hip muscle group | Measure | Intra-rater A | | | | | Intra-rater B | | | | | Interrater | | | | |
|---|---|---|---|---|---|---|---|---|---|---|---|---|---|---|---|---|
| | | Test (mean ±SD) | Retest (mean ±SD) | ICC (95% CI) | SEM (SEM %) | MDC95 (MDC %) | Test (mean ±SD) | Retest (mean ±SD) | ICC (95% CI) | SEM (SEM %) | MDC95 (MDC %) | Rater A (mean ±SD) | Rater B (mean ±SD) | ICC (95% CI) | SEM (SEM %) | MDC95 (MDC %) |
| **Flexors** | Pk | 13.11 ±6.00 | 13.32 ±6.20 | 0.931[a] (0.822–0.974) | 0.58 (4.36) | 1.60 (12.07) | 11.93 ±3.76 | 12.67 ±6.32 | 0.851[a] (0.612–0.942) | 1.04 (8.49) | 2.89 (23.52) | 13.22 ±5.90 | 12.30 ±4.85 | 0.966[a] (0.912–0.987) | 0.28 (2.22) | 0.79[b] (6.16) |
| | Af | 10.83 ±5.08 | 10.86 ±5.04 | 0.939[a] (0.841–0.976) | 0.42 (3.91) | 1.18 (10.83) | 9.54 ±2.95 | 10.24 ±4.90 | 0.761[a] (0.380–0.908) | 1.25 (12.68) | 3.47 (35.14) | 10.85 ±4.91 | 9.89 ±3.64 | 0.935[a] (0.832–0.975) | 0.42 (4.08) | 1.17 (11.31) |
| **Abductors** | Pk | 7.52 ±4.09 | 8.15 ±4.13 | 0.974[a] (0.932–0.990) | 0.17 (2.12) | 0.46[b] (5.89) | 7.87 ±4.84 | 8.21 ±4.49 | 0.927[a] (0.812–0.972) | 0.47 (5.85) | 1.30 (16.16) | 7.83 ±4.06 | 8.04 ±4.51 | 0.971[a] (0.924–0.989) | 0.18 (2.22) | 0.49[b] (6.15) |
| | Af | 5.97 ±2.99 | 6.45 ±3.15 | 0.968[a] (0.917–0.988) | 0.15 (2.42) | 0.42[b] (6.69) | 6.42 ±4.01 | 6.58 ±3.48 | 0.932[a] (0.822–0.974) | 0.35 (5.41) | 0.98 (15.01) | 6.21 ±3.02 | 6.50 ±3.63 | 0.913[a] (0.774–0.967) | 0.40 (6.28) | 1.11 (17.41) |
| **Adductors** | Pk | 9.31 ±5.05 | 10.91 ±5.69 | 0.975[a] (0.935–0.990) | 0.26 (2.60) | 0.73[b] (7.20) | 9.16 ±4.94 | 10.87 ±6.13 | 0.930[a] (0.818–0.973) | 0.57 (5.69) | 1.58 (15.77) | 10.11 ±5.32 | 9.97 ±5.33 | 0.982[a] (0.952–0.993) | 0.14 (1.36) | 0.38[b] (3.77) |
| | Af | 7.08 ±3.71 | 8.31 ±4.08 | 0.957[a] (0.888–0.983) | 0.30 (3.85) | 0.82 (10.68) | 6.95 ±3.72 | 8.38 ±4.47 | 0.945[a] (0.854–0.980) | 0.43 (5.55) | 1.18 (15.39) | 7.69 ±3.82 | 7.67 ±3.97 | 0.983[a] (0.955–0.993) | 0.09 (1.22) | 0.26[b] (3.40) |
| **Extensors** | Pk | 9.32 ±4.75 | 9.83 ±4.83 | 0.924[a] (0.797–0.972) | 0.51 (5.28) | 1.40 (14.65) | 8.64 ±4.38 | 9.97 ±4.88 | 0.940[a] (0.839–0.977) | 0.45 (4.83) | 1.25 (13.40) | 9.58 ±4.62 | 9.31 ±4.50 | 0.973[a] (0.929–0.990) | 0.17 (1.84) | 0.48[b] (5.09) |
| | Af | 7.38 ±3.64 | 7.92 ±3.76 | 0.920[a] (0.786–0.970) | 0.42 (5.46) | 1.6 (15.13) | 6.57 ±3.11 | 7.77 ±3.57 | 0.942[a] (0.844–0.978) | 0.34 (4.76) | 0.95 (13.19) | 7.65 ±3.56 | 7.17 ±3.25 | 0.957[a] (0.885–0.984) | 0.22 (2.91) | 0.60[b] (8.07) |

Pk: Peak Force (Kgf); Af: Average Force (Kgf); SD: Standard Deviation; $ICC_{2,1}$: Intra-class Correlation Coefficient; 95% CI: 95% Confidence Interval; SEM: Standard Error of Measurement; $MDC_{95}$: Minimal Detectable Change (95% CI)

[a] Good/excellent ICC ($\geq$0.75)

[b] $MDC_{95}$ < 10%.

The HHD reliability analysis demonstrated a high to very high ICC for test-retest reliability. All rater A measurements presented an excellent correlation in the test-retest analysis, considering both peak force (Pk) and average peak force (Af), while rater B presented a good ICC for flexors Pk (ICC = 0.851; 95%CI {0.612–0.942}) and flexors Af (ICC = 0.761; 95%CI {0.380–0.908}), and excellent correlations for abductors, adductors, and extensors.

The SEM ranged from 0.15 to 0.58Kgf (kilogram-force) for rater A and 0.34 to 1.25kgf for rater B, with rater A being more consistent in the test-retest measurements of Pk and Af for flexor, abductor, and adductor hip muscles. In addition, rater A obtained smaller values of MDC when considering all flexor, abductor, and adductor muscles for Pk and Af measures. This difference between raters was more pronounced in the flexors muscle group, which presented the highest mean values of strength for both raters in the test-retest measurements.

Nevertheless, when we consider the mean of the two measures in the inter-rater analysis of relative reliability, all ICCs for both Pk and Af were classified as excellent ($\geq$0.90) with good precision, expressed by the 95% CI; the smallest value was found for Abductor Af (ICC = 0.913; 95%CI{0.774–0.967}) and the highest value for Adductor Af (ICC = 0.983; 95% CI {0.955–0.993}). The absolute reliability found for Pk ranged from 0.14 to 0.28kgf, and for Af, it ranged from 0.09 to 0.42kgf, with better consistency for adductor, followed by extensor, abductor, and flexor muscle groups for both measures. These results of MDC% (95%CI) were

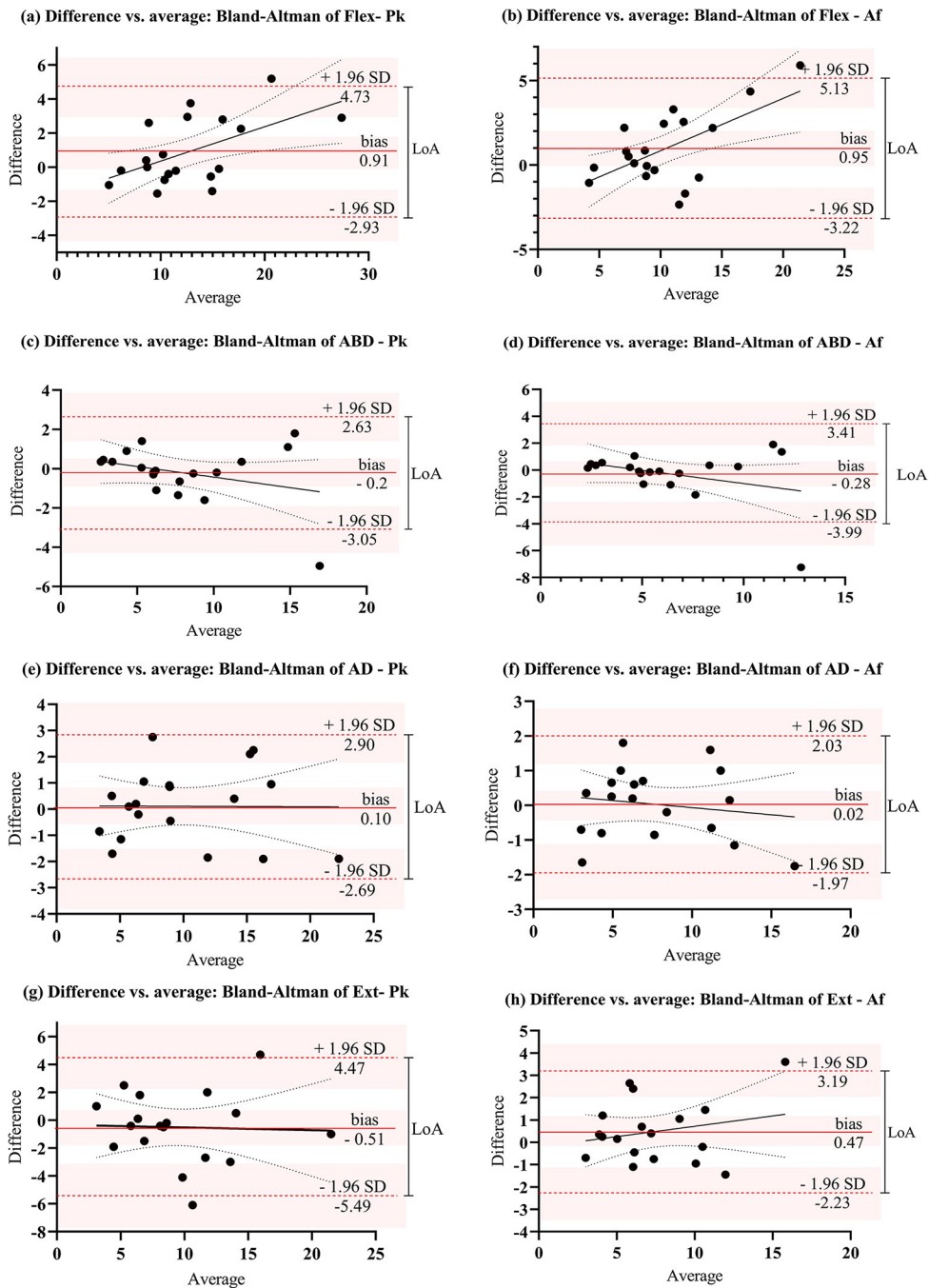

**Fig 2.** Bland-Altman plots comparing the average of all measures against the differences between the average measures (rater A-B). Each black dot represents the average of all measures (Kgf) of one individual. Dashed red lines represent the Limit of Agreement (LoA) of 95% and the continuous red line represents the bias, with their respective 95%CI (red shadow). Flex: flexors; ABD: abductors; AD: adductors; Ext: extensors; Pk: peak force; Af: Average peak force.

smaller than 10% for all Pk measures analyzed, which may reflect a satisfactory parameter when comparing the mean of two measures between different raters.

The Bland-Altman plot (Fig 2) shows the distribution of the differences in mean values between raters (A-B) versus the mean of all measures. The differences were well distributed for

abductor, adductor, and extensor muscle groups, demonstrated by the low bias for Pk and Af, with the lowest tendency of disagreement for hip adductors (Pk bias = 0.10 {LoA -2.69 to 2.90}, Fig 2E and Af bias = 0.02 {LoA -1.97 to 2.03}, Fig 2F), followed by hip abductors (Pk bias = - 0.2 {LoA -3.05 to 2.63}, Fig 2C and Af bias = -0.28 {LoA -3.99 to 3.41}, Fig 2D), and hip extensors (Pk bias = -0.51 {LoA -5.49 to 4.47}, Fig 2G and Af bias = 0.47 {LoA -2.23 to 3.19}, Fig 2H). The regression line did not show a statistically significant difference in the proportional error for those muscle groups. On the other hand, hip flexor bias demonstrated that differences in measures for rater A for Pk were, on average 0.91 Kgf higher than for rater B (Pk bias = 0.91 {LoA -2.93 to 4.73}, Fig 2A); and the differences in measures for Af were on average 0.95kgf higher than rater B (Af bias = 0.95 {LoA -3.22 to 5.13}, Fig 2B). These higher values seem to be related to a tendency of rater A to measure higher values, with increased mean flexor strength when compared to rater B. The regression analysis showed significant deviations from zero for Pk ($p = 0.01$) and Af ($p = 0.01$) in the positive direction, with a higher proportional error for rater B.

## Discussion

To our knowledge, this is the first study to assess the use of an HHD in a clinical population with symptomatic hip osteoarthritis, considering the degree of radiographic impairment and pain related to the disease. Our study was designed to reproduce a clinical situation where repeated strength measures could be collected easily in a viable routine rather than a laboratory study design. We demonstrated that the Lafayette HHD is a reliable instrument to evaluate hip muscle strength in this population, with good to excellent intra- and inter-rater reliability, satisfactory consistency, and minimal differences in the intra-rater and inter-rater analyses. Thus, clinicians can use the HHD to evaluate disuse or treatment effects on muscle strength in symptomatic hip OA patients.

Previous studies demonstrated that considering the lower limb musculature, the hip presented the strongest validity and reliability for measures of peak force, comparing the same HHD and a fixed dynamometer. Excellent reliability was also found when comparing the HHD applied by a rater or a belt system. Nevertheless, both these studies evaluated healthy and active subjects, and the authors suggest caution with generalization for the clinical population [11, 16]. Just one study assessed the HHD reliability for lower limb strength in older individuals (over 65 years old), including participants with hip and knee OA, demonstrating good intra- and inter-rater reliability for hip and knee muscle strength assessments [10]. However, only ~60% of the participants included in that study have hip or knee OA, and the descriptions of the pain and source of symptoms were poorly characterized, which makes comparisons between our results and those of Arnold and colleagues [10] difficult.

Interestingly, the present study demonstrated that the participants present good tolerance for the time taken to perform the measurements (3 seconds), even when pain was also perceived. Collectively, these data also corroborate previous results concerning older adults [10], suggesting that even when the articular disease is present in the lower limb, notably hip OA, the reliability of the HHD is satisfactory to recommend this instrument as a tool for clinical assessment. We also provide adequate information about the characteristics of the participants' hip OA, making it clear how much pain, dysfunction, and reduction in quality of life could be associated with the disease, in order to define more precisely the population of interest in this study. Despite the participants experiencing pain when performing the test protocol, the HHD test demonstrated good to excellent ICC, raising the question of the interference of patient discomfort as a potential limitation to performing tests with enough reliability, as suggested in the literature [12].

Rater A had a better correlation between test-rest measures when compared to rater B for all muscle group measurements for Pk and Af, notably in the flexors group. These results may be explained by the difference in anthropometric measurements of the raters and their presumed strength (1.80m and 85kg versus 1.69m and 68kg), demonstrated previously in the literature as a factor that could influence HHD measurements [1, 17, 41]. It is possible the use of a stabilization belt system, particularly for hip flexors, could help solve this problem, given that it does not depend on the examiner's strength [17, 42]. However, there are conflicting data in the literature regarding the advantage of belt stabilization for HHD, since this device does not provide a stabilization belt [16]. Adaptations to stabilize the device and the lack of a proper method of fixation could interfere with measurements and should be further tested and validated before any recommendations are made.

The most reliable muscle strength measurement was found for hip adductors, followed by extensors and adductors, demonstrated by excellent values of ICC and an adequate 95% CI, ranging from good to excellent reliability values. An exception was observed for intra-rater B reliability, who, despite showing good ICCs for Pk (ICC = 0.851, 95% CI {0.612–0.942}) and Af (ICC = 0.761, 95% CI {0.318–0.908}), presented a wide range of 95% CI, that could be explained by the stronger participants who had larger differences between test-retest for both raters. This result agrees with Kelln and colleagues (2008), who demonstrated that stronger muscles present wider differences in test-retest evaluations. Our data also suggest that the muscle strength assessment would be more feasible in situations with muscle weakness [11, 42, 43], expressed by the low SEM values in the inter-rater analysis.

The MDC% (95% CI) calculated in the intra-rater analysis was smaller for rater A (ranging from 5.89 to 15.13%) than for rater B, who demonstrated a much wider interval (13.19 to 35.14%). However, when using the mean of two measures for the inter-rater analysis, values of MDC% were considerably reduced, by around 8%, suggesting that at least two measurements should be taken to improve the MDC% and reduce random errors. Values under 10% are considered an adequate parameter to express any real difference instead of a random error of measurement, according to Prentice et al. (2004, quoted in Chamorro et al., 2017). Our protocol seems to be adequate for clinical purposes, since it can detect small variations that could be attributed to a real difference. Averaging two measures seems to be sufficient to reduce the variability that may result from the measuring instrument, raters, or characteristics of the measure taken, aligned with the theoretical assumption that an average score would better estimate the true value, minimizing the effect of random error [32]. This is consistent with a practical protocol of measurements that could easily be reproduced in a clinical scenario, capable of minimizing time requirements and reducing discomfort/pain from repeated strength tests in a compromised joint, such as a hip with osteoarthritis. Although MDC has been considered worthwhile to screen patient progression with good precision, future studies should consider economic evaluations of screening strategies concerning HHD assessment, with many specific challenges to overcome [44].

The Bland-Altman inter-rater analysis demonstrated small values of bias for abductors, adductors, and extensors when considering the mean of the test and retest. There was a reasonable agreement with a low bias for both variables, Pk and Af, for all muscle groups evaluated, with a tendency to a proportional error only for flexors when comparing raters. However, the LoA demonstrated a large range of fixed error, especially for flexors and extensors. Future studies should evaluate the influence of experience and routine practice on the LoA fixed error range when using this device.

Some limitations should be addressed in our study. We did not perform measures on different days and in different positions, so the conclusions raised here should be restricted to conditions that replicate this protocol and compared with caution when considering studies

performed in a different setting. With respect to raters, the experience level of both raters was the same; the inclusion of raters with different levels of expertise and practice with this instrument would reflect a more realistic scenario. The rater's ability to resist hip strength is a very relevant point that could interfere with the reproducibility of measurements [9, 12]. Considering that rater B, who weighs 68kg, had some difficulty stabilizing the HHD for hip flexor measurements, we suggest that lighter raters should be intensively trained to achieve better consistency and to rigorously follow the standardized protocol, since it is possible that knowledge of biomechanics and positioning may overcome the influence of his/her body weight and presumed strength [11, 45]. Furthermore, the sample size did not allow further analysis of the subgroup related to hip osteoarthritis classification, and the relation between radiographic impairment and HHD reliability may not be inferred from our results. Future studies are needed to evaluate the reliability of the HHD in other clinical situations, such as knee osteoarthritis.

## Conclusion

The HHD is a reliable method to evaluate hip muscle strength in individuals with symptomatic hip OA, with good to excellent intra- and inter-rater reliability and low values of SEM, even in the presence of pain related to the disease. The mean of at least two measures provides values with satisfactory agreement and reliability between raters, with adequate precision in an easily applied protocol. This study also provided values for the MDC, which could help to define a threshold to quantify improvements or reductions in hip muscle strength during treatment interventions or evaluation of disease progression with a low-cost, portable, and useful tool that requires little training for routine patient care assessment.

## Acknowledgments

We thank all participants and collaborators involved in this study.

## Author Contributions

**Conceptualization:** Gilvan Ferreira Vaz.

**Formal analysis:** Gilvan Ferreira Vaz.

**Investigation:** Gilvan Ferreira Vaz, Felipe Florêncio Freire, Marcus Alexandre Brito de Aviz.

**Methodology:** Gilvan Ferreira Vaz, Wagner Rodrigues Martins.

**Supervision:** Henrique Mansur Gonçalves, João Luiz Quagliotti Durigan.

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
