## [Decision Letter · Decision Letter 0]

16 Jan 2023

PONE-D-22-30852Intra- and inter-rater reliability, agreement, and minimal detectable change of the handheld dynamometer in individuals with symptomatic hip osteoarthritis.PLOS ONE

Dear Dr. Vaz,

Thank you for submitting your manuscript to PLOS ONE. After careful consideration, we feel that it has merit but does not fully meet PLOS ONE’s publication criteria as it currently stands. Therefore, we invite you to submit a revised version of the manuscript that addresses the points raised during the review process.

Please consider carefully the points made by both reviewers and address them accordingly. Comments by Reviewer 2, in particular, are very helpful in strengthening the practical application of the study as well as ensuring clarity on the statistical analyses used.  

We look forward to receiving your revised manuscript.

Kind regards,

Theodoros M. Bampouras

Academic Editor

PLOS ONE

Journal Requirements:

Reviewers' comments:

Reviewer's Responses to Questions

**Comments to the Author**

1. Is the manuscript technically sound, and do the data support the conclusions?

Reviewer #1: Partly

Reviewer #2: Yes

2. Has the statistical analysis been performed appropriately and rigorously? 

Reviewer #1: No

Reviewer #2: Yes

3. Have the authors made all data underlying the findings in their manuscript fully available?

Reviewer #1: Yes

Reviewer #2: Yes

4. Is the manuscript presented in an intelligible fashion and written in standard English?

Reviewer #1: Yes

Reviewer #2: Yes

5. Review Comments to the Author

Reviewer #1: It is a very interesting study. I think that it is highly significant to appropriately evaluate the hip joint muscle strength of people with hip osteoarthritis. Please correct according to comments from the editorial committee.

<major problems="">

L391-　As a result of the hip flexion, examiner B, who weighs 68 kg, may not be able to fix HHD at less than 25% of his body-weight. It is possible that this tendency may become more pronounced in examiners weighing 68 kg or less. I think there are a lot of cases where the weight of the examiner is 68 kg or less. Therefore, the reliability of hip flexion is questionable. If the examiner's body weight is even smaller, it may affect the results of other measurement items. Therefore, the weight of the examiner to whom the results of this study are applicable is considered to be limited.

L378-　The random error between examiners is around 5%, but considering that the random error within examiners is 5-20%, how should we measure in clinical practice? This is an important interpretation that will lead to clinical practice, so please add it.

<minor problems="">

L207-　Are the ICCs used in this study ICC(1,1) and (2,1)?

L217　In this study, the significance level is set to 5%, so I think it is better to use MDC95 as the MDC. The random error increases accordingly, but it is also balanced with the 95% CI of the ICC.

L299-　Although we consider systematic errors by the Bland-Altman method, please use the terms fixed error for the LOA and proportional error for the regression equation.</minor></major>

Reviewer #2: Congratulate the authors for their proposal and article. Some minor comments:

The graphics don't look good.

I recommend being consistent with the use of the point for decimals (bug in ln 292).

I recommend providing images of the tests (for example, https://doi.org/10.1080/1091367X.2020.1822363)

6. PLOS authors have the option to publish the peer review history of their article (what does this mean?). If published, this will include your full peer review and any attached files.

Reviewer #1: No

Reviewer #2: **Yes: **Rodrigo Martin-San Agustin

---

## [Author Response · Author response to Decision Letter 0]

12 Mar 2023

Dear Editors, in an effort to answer all your questions, we resubmit our manuscript with a response letter, as well as 2 files of figures, a new one (fig1) and a remade one (fig2). The answers are as follow in the word file attached.

EDITOR TEAM COMMENTS BELOW:

Reviewer #1: It is a very interesting study. I think that it is highly significant to appropriately evaluate the hip joint muscle strength of people with hip osteoarthritis. Please correct according to comments from the editorial committee.

L391-　As a result of the hip flexion, examiner B, who weighs 68 kg, may not be able to fix HHD at less than 25% of his body-weight. It is possible that this tendency may become more pronounced in examiners weighing 68 kg or less. I think there are a lot of cases where the weight of the examiner is 68 kg or less. Therefore, the reliability of hip flexion is questionable. If the examiner's body weight is even smaller, it may affect the results of other measurement items. Therefore, the weight of the examiner to whom the results of this study are applicable is considered to be limited. 

Answer: Dear reviewer, thank you very much for this important comment. The use of a handheld dynamometer is very simple, and the rater's ability to resist hip strength is a very relevant point to be highlighted, since lighter raters or even evaluations in conditions such as with healthy individuals may interfere with the reproducibility of measures. However, we partially agree with your argument, as we believe that a trained assessor, with knowledge of biomechanics and positioning, may be able to perform measurements with adequate consistency, with a minor influence of his/her body weight. To clarify this limitation, we have added in line 415 a suggestion for caution concerning extrapolation of our results to raters lighter than 68kg, presuming difficulties in resisting during the tests and consequently lower agreement between measures and raters, and suggesting intensive training to achieve better intrarater reliability.

L378 – The random error between examiners is around 5%, but considering that the random error within examiners is 5-20%, how should we measure in clinical practice? This is an important interpretation that will lead to clinical practice, so please add it.

Answer: Thank you for your helpful comment. We have tried to address the issue above in line 389, emphasizing the improvement in MDC% when considering two measures in the inter-rater analysis, suggesting that the mean of at least two measurements should be used to achieve appropriate agreement and consistency. 

L207-　Are the ICCs used in this study ICC(1,1) and (2,1)?

Answer: We appreciate the question raised, as this important information was not clear in the first version of the manuscript. We used ICC (2,1) in our statistical analysis. To make this information available in the methods, we have added information in the Statistical Analysis (line 213) and table 2 legend (line 278 and 282). 

L217-　In this study, the significance level is set to 5%, so I think it is better to use MDC95 as the MDC. The random error increases accordingly, but it is also balanced with the 95% CI of the ICC.

Answer: Thank you for your suggestion. We decided to revise the report according to your request. We had some doubts about how rigorous we should be when calculating the MDC% for manual dynamometry and its impact on clinical practice. Indeed, as expressed in table 2 (page 13), the change in CI elevated MDC and MDC% for both raters in the intra-rater analysis, mainly for rater B. However, the inter-rater comparison was still considered very satisfactory and applicable in the clinical setting. Thus, changes have been made in line 42 to point to the two variables that were not under the 10% limit; in lines 220 and 222 regarding the MDC equation; table 2 was adequate to show values with 95%CI (title, legend, and content on 7th, 12th, and 17th column); in line 305, without changes in the conclusion, and in line 389, in agreement with the alterations proposed in your second suggestion.

L299-　Although we consider systematic errors by the Bland-Altman method, please use the terms fixed error for the LOA and proportional error for the regression equation.

Answer: Dear reviewer, in an effort to use the correct terms considering the Bland-Altman method, we have made some changes in lines 316, 322, 406, 407, and 409.  

Reviewer #2: Congratulate the authors for their proposal and article. Some minor comments:

1) The graphics don't look good.

Answer: Dear reviewer, thank you for your attention regarding the graphics. We have reformulated them to give readers clearer and more precise information, including the 95%CI for bias and LoA, according to the formula provided by Giavarina et al (2015). In addition, we have improved the quality of the image. The figure is resubmitted with alterations. 

Reference: Giavarina D. Understanding Bland Altman analysis. Biochem Med (Zagreb) 2015; 25:141–51. https://doi.org/10.11613/BM.2015.015.

2) I recommend being consistent with the use of the point for decimals (bug in ln 292).

Answer: Thank you very much for the comment. We have carefully reviewed the manuscript, and changes have been made in lines 301 and 302, to correct the use of the point for decimal numbers. 

3) I recommend providing images of the tests for example (https://doi.org/10.1080/1091367X.2020.1822363):

Answer: Thank you again for your recommendation. We appreciate your thoughtful suggestion. To illustrate the protocol sequence we used, we have added a figure composed of four parts, to allow readers to visualize the position of the rater and participant during the test routine, named Fig1 (A-D). As a consequence, the corresponding legend has been added in line 181, references to this figure have been added in lines 161, 167, 171, and 175, and the title and legend of the graphics have been renamed as figure 2.

---

## [Decision Letter · Decision Letter 1]

29 Mar 2023

PONE-D-22-30852R1Intra- and inter-rater reliability, agreement, and minimal detectable change of the handheld dynamometer in individuals with symptomatic hip osteoarthritis.PLOS ONE Dear Dr. Vaz,

Thank you for submitting your manuscript to PLOS ONE. After careful consideration, we feel that it has merit but does not fully meet PLOS ONE’s publication criteria as it currently stands. Therefore, we invite you to submit a revised version of the manuscript that addresses the points raised during the review process.

You will see that the reviewers were satisfied by your replies. Please address these final, specific (minor) points below, before the manuscript is accepted for publication:

The reply to Reviewer 1's point regarding raters of <68kg mass, is more informative than what was included in the manuscript. Please expand in the manuscript on this limitation in line with your response, as it is a crucial point.You discuss the reduction in error by averaging the two trials and state that this makes the measurement sufficiently sensitive to detect clinically meaningful changes, but provide no reference / reminder of what those changes are (and thus that the averaged measurements error allows detection of  such changes). Reference 3 might be sufficient - but you need to provide a 'yardstick' for that statement. The statement in Lines 273-275 re the increased pain not affecting the measurement can only hold true if you knew that the participants truly performed at maximum capacity; however, this is not possible to know. For example, it could be that the increased pain in the second trial resulted in lower force generation (e.g. https://ddec1-0-en-ctp.trendmicro.com:443/wis/clicktime/v1/query?url=https%3a%2f%2fwww.frontiersin.org%2farticles%2f10.3389%2ffpsyg.2010.00210%2ffull&umid=d55a5243-5760-49d2-acea-601c4247b06a&auth=6b639a990a359ff1d6cc8761081d57748ce3c81e-650e099fe74b65ee9f9331a4049e4dd2d0c53860) and, therefore, a closer score to the first trial which would, in turn, result in higher ICC but through a submaximal second trial and a maximal first trial. On the other hand, it could be that the difference of <1 cm in pain is not clinically significant / meaningful (e.g. https://ddec1-0-en-ctp.trendmicro.com:443/wis/clicktime/v1/query?url=https%3a%2f%2fpubmed.ncbi.nlm.nih.gov%2f9428860%2f&umid=d55a5243-5760-49d2-acea-601c4247b06a&auth=6b639a990a359ff1d6cc8761081d57748ce3c81e-80734d227bdb7133929fdb958254a41191226b88). Please revise that point to remove the statement you currently have. Please submit your revised manuscript by May 13 2023 11:59PM. If you will need more time than this to complete your revisions, please reply to this message or contact the journal office at plosone@plos.org. Please include the following items when submitting your revised manuscript:A rebuttal letter that responds to each point raised by the academic editor and reviewer(s). You should upload this letter as a separate file labeled 'Response to Reviewers'.A marked-up copy of your manuscript that highlights changes made to the original version. You should upload this as a separate file labeled 'Revised Manuscript with Track Changes'.An unmarked version of your revised paper without tracked changes. You should upload this as a separate file labeled 'Manuscript'.If applicable, we recommend that you deposit your laboratory protocols in protocols.io to enhance the reproducibility of your results. Protocols.io assigns your protocol its own identifier (DOI) so that it can be cited independently in the future. For instructions see: https://journals.plos.org/plosone/s/submission-guidelines#loc-laboratory-protocols. Additionally, PLOS ONE offers an option for publishing peer-reviewed Lab Protocol articles, which describe protocols hosted on protocols.io. Read more information on sharing protocols at https://plos.org/protocols?utm_medium=editorial-email&utm_source=authorletters&utm_campaign=protocols.

We look forward to receiving your revised manuscript.

Kind regards,

Theodoros M. Bampouras

Academic Editor

PLOS ONE

Journal Requirements:

Reviewers' comments:

Reviewer's Responses to Questions

**Comments to the Author**

1. If the authors have adequately addressed your comments raised in a previous round of review and you feel that this manuscript is now acceptable for publication, you may indicate that here to bypass the “Comments to the Author” section, enter your conflict of interest statement in the “Confidential to Editor” section, and submit your "Accept" recommendation.

Reviewer #1: All comments have been addressed

Reviewer #2: All comments have been addressed

2. Is the manuscript technically sound, and do the data support the conclusions?

Reviewer #1: Yes

Reviewer #2: Yes

3. Has the statistical analysis been performed appropriately and rigorously? 

Reviewer #1: Yes

Reviewer #2: Yes

4. Have the authors made all data underlying the findings in their manuscript fully available?

Reviewer #1: Yes

Reviewer #2: Yes

5. Is the manuscript presented in an intelligible fashion and written in standard English?

Reviewer #1: Yes

Reviewer #2: Yes

6. Review Comments to the Author

Reviewer #1: The resubmitted manuscript responds appropriately to comments. However, Fig.1 and 2 could not be confirmed because they were not attached. In the rehabilitation of hip osteoarthritis, I think that a measurement that can clearly indicate hip joint muscle strength is very effective. I hope that this kind of muscle strength measurement will spread in clinical practice.

Reviewer #2: (No Response)

7. PLOS authors have the option to publish the peer review history of their article (what does this mean?). If published, this will include your full peer review and any attached files.

Reviewer #1: No

Reviewer #2: No

---

## [Author Response · Author response to Decision Letter 1]

8 May 2023

EDITOR TEAM COMMENTS BELOW:

Reviewer’s comment #1: The reply to Reviewer 1's point regarding raters of <68kg mass, is more informative than what was included in the manuscript. Please expand in the manuscript on this limitation in line with your response, as it is a crucial point.

Answer: Dear reviewer, thank you for pointing out that the information in the manuscript was not sufficiently clear in our previous response letter. We have now added a new statement regarding the limitations of this paper in lines 421-427 aligned with our previous response, as requested. 

We have expanded our previous statement from: “Considering that rater B, who weighs 68kg, had some difficulty stabilizing the HHD for hip flexor measurements, we suggest that lighter raters should be intensively trained to achieve better intra-rater reliability” to the following phrase: “The rater's ability to resist hip strength is a very relevant point that could interfere with the reproducibility of measurements [1,2]. Considering that rater B, who weighs 68kg, had some difficulty stabilizing the HHD for hip flexor measurements, we suggest that lighter raters should be intensively trained to achieve better consistency and to rigorously follow the standardized protocol, since it is possible that knowledge of biomechanics and positioning may overcome the influence of his/her body weight and presumed strength [3]”.

References: 

1- Kelln BM, McKeon PO, Gontkof LM, Hertel J. Hand-held dynamometry: reliability of lower extremity muscle testing in healthy, physically active,young adults. J Sport Rehabil 2008;17:160–70. https://doi.org/10.1123/jsr.17.2.160.

2- Krause DA, Neuger MD, Lambert KA, Johnson AE, DeVinny HA, Hollman JH. Effects of examiner strength on reliability of hip-strength testing using a handheld dynamometer. J Sport Rehabil 2014;23:56–64. https://doi.org/10.1123/jsr.2012-0070.

3- Morin M, Hébert LJ, Perron M, Petitclerc É, Lake SR, Duchesne E. Psychometric properties of a standardized protocol of muscle strength assessment by hand-held dynamometry in healthy adults: a reliability study. BMC Musculoskelet Disord. 2023 Apr 14;24(1):294. doi: 10.1186/s12891-023-06400-2. PMID: 37060020; PMCID: PMC10103411.

Reviewer’s comment #2: You discuss the reduction in error by averaging the two trials and state that this makes the measurement sufficiently sensitive to detect clinically meaningful changes, but provide no reference/reminder of what those changes are (and thus that the averaged measurements error allows detection of such changes). Reference 3 might be sufficient - but you need to provide a 'yardstick' for that statement. 

Answer: 

Dear reviewer, we have expanded our discussion by adding new references to anchor our discussion concerning the number or measurements, as stated in lines 397-401. Our results suggest that the protocol of measurement with HHD is reliable. We consider the mean of two measures, as the results expressed higher values of ICC when comparing raters. Sources of variability may result from the measuring instrument, raters, or characteristics of the measure taken [1,2]. Biological variations in generating peak force, such as muscle fibers, elicited fatigue, limb and device positioning, stabilization of the HHD, attention of subject/raters, environmental conditions and symptoms related to the disease, and others, may influence the measurements taken. When averaging results across trials, the sources of variability between these measurements are theoretically reduced, as stated by Portney and Watkins (2015). The authors described that random error is not related to the true score when enough measurements are taken. Positive random errors would eventually cancel negative ones, making the average scores a reasonable estimate of the true score. Therefore, averaging two measures would help to reduce variability and consequently improve ICC values [1]. This is consistent with a practical protocol of measurements that could easily be reproduced in a clinical scenario. It can also minimize the time requirements and reduce discomfort/pain from repeated strength tests in a compromised joint, such as a hip with osteoarthritis. 

Regarding what was described in the discussion section: “seems to be reliable for clinical purposes since it can detect small variations that could be attributed to a real clinical change”; We have rephrased this statement, excluding the word “clinical”, to make sure that readers do not mistake it with the minimal clinical important difference (MCID) [3,4], since we refer to the minimal detectable change (MDC) that could be attributed to a real difference of peak force and average peak force instead of a difference resulting from random error. 

References:

1- Portney, Leslie Gross MPW. Foundations of Clinical Research: Applications to Practice. 3rd ed. Upper Saddle River, New Jersey: Pearson/Prentice Hall; 2015.

2- Bialocerkowski, A. E., & Bragge, P. (2008). Measurement error and reliability testing: Application to rehabilitation. International Journal of Therapy and Rehabilitation, 15(10), 422–427. https://doi.org/10.12968/ijtr.2008.15.10.31210

3- Suijker, J. J., Van Rijn, M., Ter Riet, G., van Charante, E. M., De Rooij, S. E., & Buurman, B. M. (2017). Minimal important change and minimal detectable change in activities of daily living in community-living older people. The journal of nutrition, health & aging, 21, 165-172.

4- Turner, D., Schünemann, H. J., Griffith, L. E., Beaton, D. E., Griffiths, A. M., Critch, J. N., & Guyatt, G. H. (2010). The minimal detectable change cannot reliably replace the minimal important difference. Journal of clinical epidemiology, 63(1), 28-36.

Reviewer’s comment #3: The statement in Lines 273-275 are the increased pain not affecting the measurement can only hold true if you knew that the participants truly performed at maximum capacity; however, this is not possible to know. For example, it could be that the increased pain in the second trial resulted in lower force generation (e.g. https://ddec1-0-en-ctp.trendmicro.com:443/wis/clicktime/v1/query?url=https%3a%2f%2fwww.frontiersin.org%2farticles%2f10.3389%2ffpsyg.2010.00210%2ffull&umid=d55a5243-5760-49d2-acea-601c4247b06a&auth=6b639a990a359ff1d6cc8761081d57748ce3c81e-650e099fe74b65ee9f9331a4049e4dd2d0c53860) and, therefore, a closer score to the first trial which would, in turn, result in higher ICC but through a submaximal second trial and a maximal first trial. On the other hand, it could be that the difference of <1 cm in pain is not clinically significant / meaningful (e.g. https://ddec1-0-en-ctp.trendmicro.com:443/wis/clicktime/v1/query?url=https%3a%2f%2fpubmed.ncbi.nlm.nih.gov%2f9428860%2f&umid=d55a5243-5760-49d2-acea-601c4247b06a&auth=6b639a990a359ff1d6cc8761081d57748ce3c81e-80734d227bdb7133929fdb958254a41191226b88). Please revise that point to remove the statement you currently have. 

Answer: We would like firstly to thank you for the references suggested. They were well-conducted studies and essential to discuss our data. We have performed a modification following your comment. We searched the literature to verify if, beyond the imprecision of VAS for pain around ± 20mm as pointed out by DeLoach et al. (1998), there was some definition of MCID concerning our study population. We found that a clinical perceived difference in individuals with hip osteoarthritis depends on the baseline value. Therefore, if we consider the first trial as a baseline, values between the interval of 50-65 would require a difference of 24mm to be considered clinically important, and for baseline values > 65mm, a clinically perceived difference must be even higher, 30mm. Thus, the statistical difference between the two trials of rater A was not high enough to be considered clinically relevant. Comparing the first and second trials of rater B, differences were neither statistically significant nor clinically meaningful, similar to the interrater comparison. In conclusion, besides differences that were not clinically relevant, the test protocol was still painful, and statements regarding the influence of pain on the capability of generating maximal force in all trials and, consequently the instrument reliability could not be inferred in this study design. We have removed the statement: “Although there was an existing difference in VAS for pain intensity after the rater A test compared to the retest, it did not seem to have a relevant effect on intra-rater ICC, since rater A presented better ICC and lower SEM values than rater B”; and added some comments in line 273-277 according to your suggestion.

References: 

1- Stauffer ME, Taylor SD, Watson DJ, Peloso PM, Morrison A. Definition of Nonresponse to Analgesic Treatment of Arthritic Pain: An Analytical Literature Review of the Smallest Detectable Difference, the Minimal Detectable Change, and the Minimal Clinically Important Difference on the Pain Visual Analog Scale. Int J Inflam 2011; 2011:1–6. https://doi.org/10.4061/2011/231926.

2- Tubach F, Ravaud P, Baron G, Falissard B, Logeart I, Bellamy N, et al. Evaluation of clinically relevant changes in patient reported outcomes in knee and hip osteoarthritis: the minimal clinically important improvement. Ann Rheum Dis 2005; 64:29–33. https://doi.org/10.1136/ard.2004.022905.

---

## [Editor Report · Decision Letter 2]

10 May 2023

Intra- and inter-rater reliability, agreement, and minimal detectable change of the handheld dynamometer in individuals with symptomatic hip osteoarthritis.

PONE-D-22-30852R2

Dear Dr. Vaz,

Thank you for the very thoughtful addressing of the relevant points. We’re pleased to inform you that your manuscript has been judged scientifically suitable for publication and will be formally accepted for publication once it meets all outstanding technical requirements.

Kind regards,

Theodoros M. Bampouras

Academic Editor

PLOS ONE
---

## [Editor Report · Acceptance letter]

1 Jun 2023

PONE-D-22-30852R2 

Intra- and inter-rater reliability, agreement, and minimal detectable change of the handheld dynamometer in individuals with symptomatic hip osteoarthritis. 

Dear Dr. Vaz:

I'm pleased to inform you that your manuscript has been deemed suitable for publication in PLOS ONE. Congratulations! Your manuscript is now with our production department. 

Kind regards, 

on behalf of

Dr. Theodoros M. Bampouras 

Academic Editor

PLOS ONE